ெ | Physiology and Metabolism | Research Article

# Nutrient conditions affect antimicrobial pharmacodynamics in *Pseudomonas aeruginosa*

Maik Kok,[1] Thomas Hankemeier,[1] J. G. Coen van Hasselt[1]

**ABSTRACT** The infectious microenvironment in chronic respiratory tract infections is characterized by substantial variability in nutrient conditions, which may impact colonization and treatment response of pathogens. Metabolic adaptation of the cystic fibrosis (CF)-associated pathogen *Pseudomonas aeruginosa* has been shown to lead to changes in antibiotic sensitivity. The impact of specific nutrients on the response to antibiotics is, however, poorly characterized. Here, we investigated how different carbon sources impact the antimicrobial pharmacodynamic responses in *P. aeruginosa*. We evaluated the effect of six antibiotics (aztreonam, ceftazidime, ciprofloxacin, colistin, imipenem, and tobramycin) on *P. aeruginosa* cultured in a basal medium enriched for seven different carbon sources (alanine, arginine, aspartate, glucose, glutamate, lactate, and proline). Pharmacodynamic responses were characterized by measuring time-kill profiles for a bioluminescent *P. aeruginosa* PAO1 *Xen41* strain. We show that single-nutrient modifications minimally affected bacterial growth rate. For specific nutrient-antibiotic combinations, we find relevant alterations in antibiotic sensitivity (i.e., $EC_{50}$) and the maximum drug effect ($E_{max}$), in particular for ciprofloxacin, colistin, imipenem, and tobramycin. The most pronounced effect was observed for tobramycin, where glucose was found to reduce the $EC_{50}$ (0.5-fold), whereas lactate-enriched conditions led to a 4.3-fold increase in $EC_{50}$. Using pharmacokinetic-pharmacodynamic simulations, we illustrate that the magnitude of the nutrient-driven pharmacodynamic changes impact treatment for clinical dosing strategies of tobramycin. In summary, this study underscores the impact of nutrient composition on antimicrobial pharmacodynamics, which could potentially contribute to observed variability of antimicrobial treatment responses in CF patients.

**IMPORTANCE** Chronic respiratory tract infections in cystic fibrosis patients present significant challenges for antibiotic treatment due to the complexity of the respiratory environment. This study investigated how variations in nutrient levels, altered during chronic infections, affect pathogen response to antibiotics in an experimental setting. By simulating different nutrient conditions, we aimed to uncover interactions between nutrient availability and antibiotic sensitivity. Our findings provide critical insights that could lead to more effective treatment strategies for managing chronic respiratory tract infections in cystic fibrosis patients while also guiding future research in improving treatment methodologies.

**KEYWORDS** antibiotics, cystic fibrosis, *Pseudomonas aeruginosa*, nutrients

**Peer Reviewer** Cheryl L. Quinn

Address correspondence to J. G. Coen van Hasselt, coen.vanhasselt@lacdr.leidenuniv.nl.

The authors declare no conflict of interest.

Cystic fibrosis (CF)-associated lung infections are facilitated by a complex infectious microenvironment involving a dense mucus layer harboring a diverse array of potential microbial nutrients (1). Antibiotic treatment in patients with CF often yields unpredictable outcomes and aligns poorly with routine antimicrobial susceptibility testing (2, 3). Profound variability in microbial nutrients is observed within the chronic

infectious environment, both across and within patients (4, 5). Unlike many other bacterial pathogens, the pathogen *Pseudomonas aeruginosa* prioritizes the utilization of a wide array of carbon sources over glucose, including alanine, arginine, aspartate, glutamate, proline, and lactate (6, 7). This metabolic versatility may explain its pervasive presence in chronic CF-associated infections and provide a competitive advantage during antibiotic treatment (8–10).

Alterations in metabolic processes associated with differences in available nutrients may impact the response to antibiotic treatment in *P. aeruginosa* (11–13). For example, nutrient deprivation prevents cell wall modifications due to its high energy demand, enhancing the effect of cell wall targeting antibiotics (e.g., polymyxins and β-lactams) (14–16). The supplementation of metabolites to activate energy production through aerobic respiration in nutrient-deprived environments can increase sensitivity toward fluoroquinolones and aminoglycosides (17–19). Although these changes illustrate the modulatory role of deprived nutrient conditions and microbial metabolism on the response to antibiotics, insights into the contribution of nutrients relevant to CF lung microenvironments remain limited.

To assess the effects of nutrient conditions on antimicrobial pharmacodynamics (PD), conventional readouts such as the minimum inhibitory concentrations (MIC) have important limitations, as this is a static composite measure. More comprehensive characterization of changes in the pharmacodynamic response to antibiotics can be achieved through time-kill studies, which monitor bacterial densities over time when exposed to antibiotics, allowing the evaluation of bacterial growth, antibiotics-associated killing, and adaptation effects (20, 21). Although time-kill studies provide these valuable insights, they remain limited in their throughput and the number of time points at which data can be collected (22). The use of bacterial strains carrying luminescent reporters allows real-time monitoring of bacterial growth and killing dynamics during antibiotic exposure (23, 24). The resulting profiles can be analyzed using mathematical pharmacodynamic models to obtain further quantitative insights into PD relationships. As such, the use of luminescence-based time-kill studies in combination with quantitative pharmacodynamic models is well-suited for comprehensively assessing the effects of nutrient conditions on antibiotic response.

In the current study, we aimed to systematically evaluate the impact of a wide range of CF sputum-relevant carbon sources on antimicrobial time-kill responses in *P. aeruginosa*. The nutrients evaluated included alanine, arginine, aspartate, glutamate, lactate, proline, and glucose. These nutrient-associated effects were evaluated for six antibiotics commonly used for respiratory tract infections in CF, including aztreonam, ceftazidime, ciprofloxacin, colistin, imipenem, and tobramycin. We assessed the bacterial growth/kill time course profiles using extensive time-kill studies with a modified *P. aeruginosa* PAO1 strain carrying a constitutively active luminescent reporter. This strain was subsequently used to infer PD parameters and perform pharmacokinetic-pharmacodynamic (PK-PD) simulations to demonstrate the potential clinical impact of nutrients on antimicrobial PD.

## MATERIALS AND METHODS

### Culture media and bacterial strain

A basal medium was prepared consisting of physiologically relevant concentrations of amino acids in synthetic CF sputum as described previously (7), calcium and magnesium adjusted 0.11 M phosphate buffer, ammonium chloride, potassium nitrate, ferrous sulfate, Basal Medium Eagle 1× vitamins, and trace metals. The pH of the basal medium was confirmed to be 7.4 and was verified after the addition of nutrients and filter sterilization. The specific concentrations of all medium components are listed in Table S1. We then prepared seven unique nutrient-specific media for each of the carbon sources used in this study, including alanine, arginine, aspartate, glutamate, glucose, proline, and lactate. Each of these nutrients was added separately to the basal medium in excess at

a concentration of 15 mM. The *P. aeruginosa* bioluminescent strain PAO1 Xen41 (Revvity Inc., Waltham, MA, USA) was used in all experiments. The promoterless insertion of the *luxCDABE* cassette into the chromosomal genome resulted in a linear relationship between luminescence in relative light units (RLU) and CFU/mL (Fig. S1) (23, 24).

## Antibiotics

Antibiotic stock solutions were freshly prepared on the day of the experiment and diluted to the desired concentrations using an Opentrons OT-2 (Opentrons Inc., New York, NY, USA) liquid handling system. Aztreonam and ceftazidime pentahydrate were purchased from Thermo Fisher Scientific (Breda, the Netherlands). Ciprofloxacin, imipenem monohydrate, and tobramycin were purchased from Chem-Impex International (Wood Dale, IL, USA). Colistin sulfate was purchased from Cayman Chemical Company (Ann Arbor, MI, USA).

## Experimental workflow

Time-kill assays were conducted by culturing *P. aeruginosa* in each of the nutrient-specific media formulations and exposing the cultures to six different antibiotics. We tested nine different serially diluted concentrations in a microtiter plate format, centered around their minimal inhibitory concentrations (Fig. 1). All experiments were conducted at 37°C and with shaking at 150 rpm.

The PAO1 Xen41 strain was streaked on LB agar plates and incubated overnight. One colony was transferred to a nutrient-specific media formulation (4 mL) and cultured overnight. The liquid cultures were diluted to an optical density at 600 nm (OD$_{600}$) of 0.05 before inoculation, corresponding to an approximate bacterial concentration of $5*10^6$ CFU/mL. The bacterial inoculum (50 µL) was added to a fresh medium with antibiotics (150 µL) in a white 96-well microtiter plate.

After inoculation, microtiter plates were transferred to a Liconic StoreX STX44 incubator (Mauren, Principality of Liechtenstein) for incubation (95% relative humidity). A Peak Analysis and Automation KX-2 Laboratory Robot (Hampshire, UK) transferred the microtiter plate every hour between the incubator and the BMG Labtech Fluostar Omega microplate reader (Ortenberg, Germany) for time-course data acquisition. The density of viable bacteria was determined by measuring luminescence and quantified as relative light units (RLU).

## Data processing and analysis

All data preprocessing and analyses are performed using R. To evaluate fitness differences between growth media, the maximal population growth rates (µ$_{max}$) and the maximal population density (N$_{max}$) under antibiotic-free culture conditions were calculated using the all splines function from the grofit package (25). Differences in growth parameters in the studied media formulations compared with the basal media were assessed by the Dunnett's Test from the DescTools R package (26).

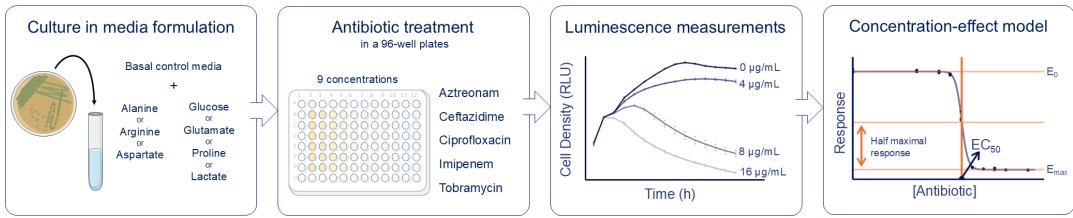

**FIG 1** Experimental approach. The experiment started with a liquid culture in the media formulation containing one or none of the nutrients of interest. The population was diluted to the starting density and treated with nine concentrations of antibiotic, whereas the luminescence was determined every hour in relative light units (RLU). A four-parameter log-logistic function was fitted on the area under the curve or growth rate per antibiotic concentration to determine the upper limit (E$_0$), lower limit (E$_{max}$), and half-maximal effective concentration (EC$_{50}$).

To quantify drug effects, the total bacterial burden was determined by calculating the area under the curve (AUC) of the RLU between 1 and 15 h of incubation (Fig. S2). The resulting AUC values were then used to quantify pharmacodynamic parameters. We fitted for each antibiotic-nutrient combination the mean ($n = 3$) AUC to the antibiotic concentration ([AB]) using a four parameter log-logistic (LL.4) function from the drc R package (Equation 1) (27). This function includes parameters for the hill coefficient ($n_H$), the lower limit ($E_{max}$), the upper limit ($E_0$), and the relative half-maximal effective concentration ($EC_{50}$). The difference in relative $EC_{50}$ among culture conditions was quantified using the 95% CI.

$$AUC([AB]) = E_{max} + \frac{E_0 - E_{max}}{1 + e^{n_H(\log([AB]) - \log(EC_{50}))}} \tag{1}$$

## Pharmacokinetic-pharmacodynamic (PK-PD) simulations

We used a previously published pharmacokinetic (PK) model for tobramycin to perform PK-PD simulations (28). We simulated the clinical concentration-time profiles for a typical dose of 3.3 mg/kg of intravenous tobramycin, administered every 8 h (Table S2). Interpatient variability for the parameters was derived from published interquartile ranges. Antibiotic PD was described by first estimating growth/kill rates for each antibiotic concentration, which were subsequently fitted to a pharmacodynamic sigmoidal function relating antibiotic growth/kill rate to antibiotic concentration. The growth rates were determined by determining the slope of the phase of the luminescence time kill curve where the drug effect occurred (Fig. S6), using the grofit package.

## RESULTS

### Nutrient-dependent shift in antibiotic sensitivity

We cultured *P. aeruginosa* under various nutrient conditions in the presence of different antibiotics to investigate the effect of nutrients on the pharmacodynamic (PD) response. To summarize the bacterial response kinetics–encompassing growth enhancement, suppression, or killing during antibiotic treatment, we calculated the AUC of the luminescence time course profiles. We then regressed the AUC values against antibiotic concentrations using a sigmoidal Emax model, allowing one to visualize differences in the pharmacodynamic response across conditions (Fig. 2). Overall, these analyses revealed significant effects of nutrients on the antibiotic concentration required to achieve 50% of the total antimicrobial effect (relative $EC_{50}$) and the steepness of the concentration-response profiles (Fig. S3).

The relative $EC_{50}$ would be the primary metric of relevance to quantitatively indicate subtle changes in drug potency, that is, antibiotic sensitivity across conditions. For several nutrient conditions, we observed clinically relevant alterations in the $EC_{50}$ values across different antibiotics (Fig. 3A). We observed both reductions in $EC_{50}$ as compared with the basal media and increased $EC_{50}$ values, indicating increased resistance. Across all antibiotics, no clear trends in $EC_{50}$ shifts were observed for specific nutrients.

When comparing the relative change in $EC_{50}$ with the basal medium (Fig. 3B), both aztreonam and ceftazidime exhibited similarly enhanced sensitivity across different nutrient conditions. The most notable changes were the increased sensitivity observed in lactate-enriched media for both antibiotics. In contrast, imipenem sensitivity was consistently reduced in all nutrient-enriched conditions, with the most significant reductions observed in aspartate- and glutamate-enriched media.

For ciprofloxacin, colistin, and tobramycin, a wider variation in effect was compared with the basal medium. Glucose- and proline-enriched media resulted in a reduction in $EC_{50}$, whereas aspartate-, glutamate-, and lactate-enriched media increased the $EC_{50}$ for all three antibiotics. The largest change in sensitivity was observed for tobramycin, where for lactate-rich media, the $EC_{50}$ value increased profoundly (log2(FC_EC50) = 2.09, a 4.4-fold increase).

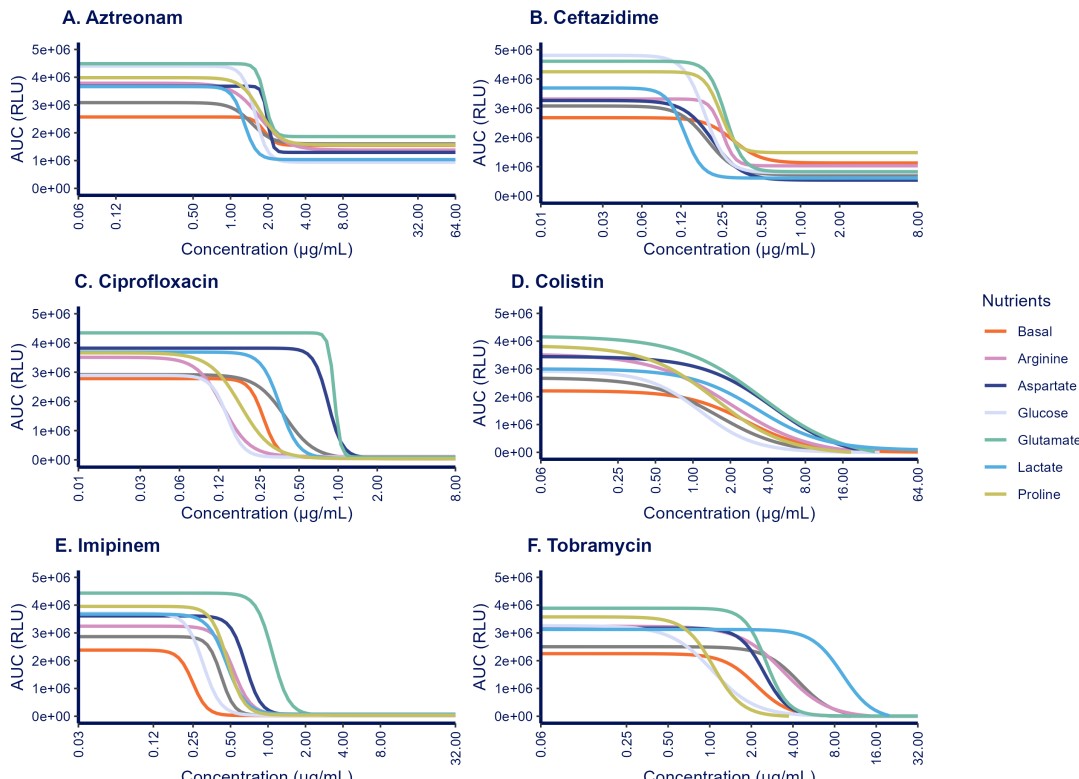

**FIG 2** Pharmacodynamic exposure-response relationships for antibiotics cultured under different nutrient conditions. The area under the curve (AUC) for bacterial growth/kill based on relative light units (RLU) up to 15 h in relation to antibiotic concentrations ($n = 9$) was fitted using sigmoidal $E_{max}$ curves, for different nutrient-enriched media formulations and the basal control media conditions. The lines represent the mean predictions derived from three biological replicates ($n = 3$).

## Fitness differences in different culture conditions affect PD parameters

We studied the effect of different nutrient-enriched media under antibiotic-free conditions on fitness and growth yield using the growth curve profiles (Fig. 4) to understand their potential contributions to differences in antibiotic response. Except for alanine, for all nutrients, we found an increase of >1.5-fold in the upper limit of the model ($E_0$), that is, the antibiotic baseline with no antimicrobial effect used in our pharmacodynamic analyses (Fig. S3). To further understand these effects, we calculated the maximum population growth rate ($\mu_{max}$) and the maximum population density ($N_{max}$) of antibiotic-free conditions (Fig. S4). Although the nutrient composition significantly affected $\mu_{max}$, the magnitude of the effect was modest (Fig. 4B), with an increase of up to 1.2-fold compared with the basal control media observed only for aspartate and glutamate. The observed effects on $E_0$ are predominantly explained by differences in $N_{max}$ (Fig. 4C), with a > 2-fold increase observed for aspartate and glutamate and a fold change between 1.2 and 2.0 for all other nutrient conditions. Distinct differences in growth curves during the transition from the exponential growth phase to the stationary phase were visible (Fig.S4), in particular for the time required to reach $N_{max}$ ($t_{max}$).

The impact of differences in $E_0$ across different nutrient conditions on PD parameters was further evaluated by analyzing the total antimicrobial response. Comparing the relative $EC_{50}$ with the absolute $EC_{50}$ provides an indication of how the limits of PD model influence the total antimicrobial effect. The relative $EC_{50}$ is defined as the midpoint between the two limits of concentration-response curve, whereas the absolute $EC_{50}$ denotes a 50% reduction in the AUC from the baseline with no antimicrobial effect ($E_0$). A larger discrepancy between these $EC_{50}$ values suggests a stronger impact of the two limits on determining the antibiotic $EC_{50}$ (29). For treatments with ciprofloxacin, colistin,

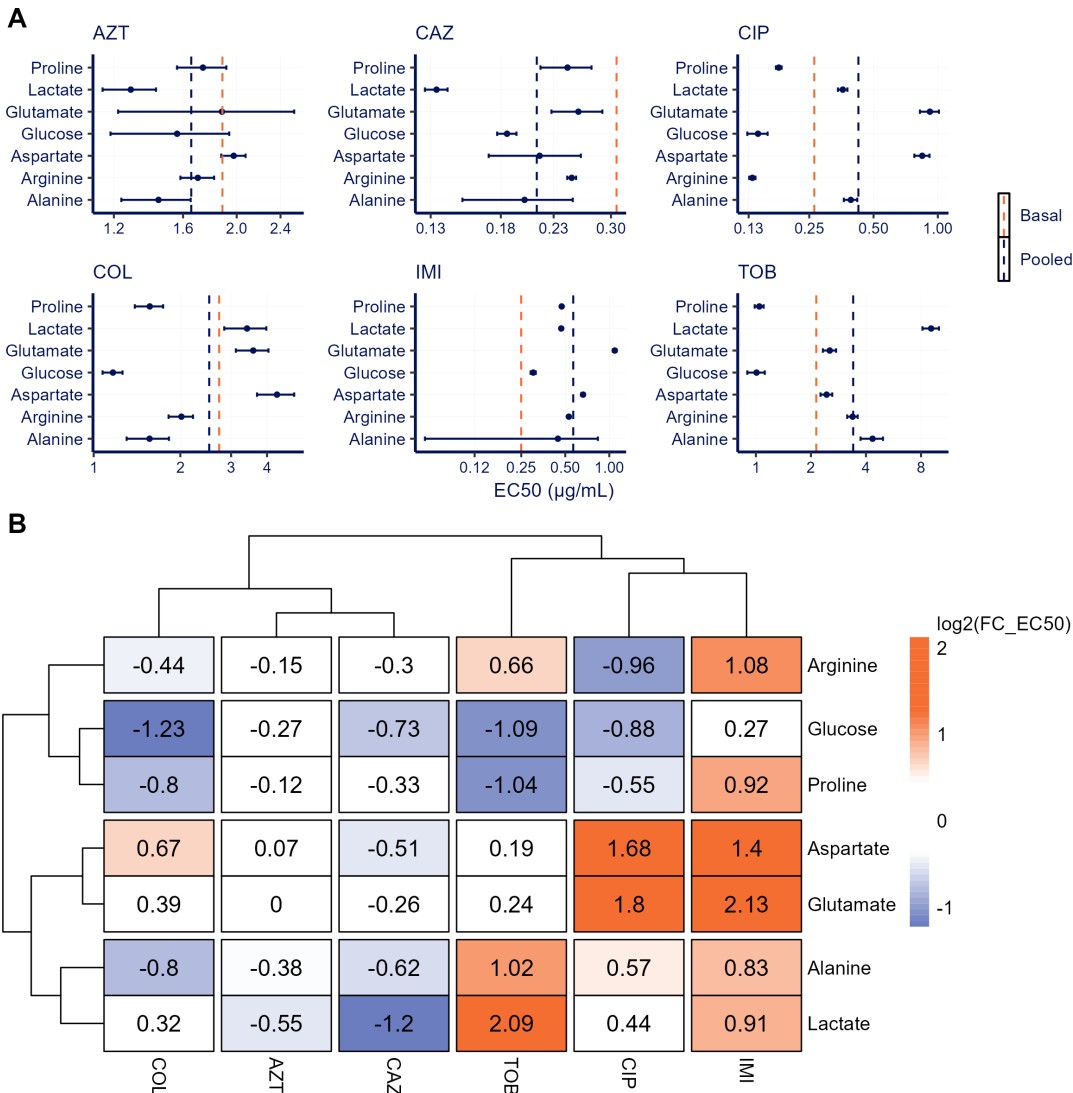

**FIG 3** Changes in antibiotic sensitivity ($EC_{50}$) of *P. aeruginosa* across different nutrients and antibiotics. Observed areas under the curve for bacterial growth and kill for *P. aeruginosa* PAO1-Xen41 were regressed against drug concentrations for different antibiotics and nutrients, using a sigmoidal Emax function. The resulting $EC_{50}$ estimates for different antibiotic-nutrient combinations are shown for (A) absolute EC50 values (mean and 95% confidence intervals), with vertical dashed lines indicating the EC50 obtained from the base media control treatment, and the cross-nutrient median $EC_{50}$, and (B) median fold-change (FC) values in $EC_{50}$, compared with the base media $EC_{50}$. The antibiotics and nutrients were clustered using Euclidean distance clustering to showcase patterns of antibiotic sensitivity and nutrient effect. Abbreviations: aztreonam (AZT), ceftazidime (CAZ), ciprofloxacin (CIP), colistin (COL), imipenem (IMI), and tobramycin (TOB).

imipenem, and tobramycin, the difference between the average relative and absolute $EC_{50}$ values was less than 5% (Fig. S5). In contrast, ceftazidime and aztreonam treatments showed a difference of, respectively, 14% and 22%, indicating that differences in the PD model limits between the nutrient conditions do influence the determination of $EC_{50}$.

## *In vitro* nutrient-driven PD differences impact treatment simulations with a clinically relevant tobramycin PK profile

To assess whether the magnitude of nutrient-associated changes in the PD response observed *in vitro* may have significance at clinically relevant antibiotic concentrations, we performed pharmacokinetic-pharmacodynamic (PK-PD) simulations. For proof of concept, we focused on tobramycin and the nutrients glucose and lactate, since for this antibiotic and these nutrient conditions clearly divergent PD effects were observed.

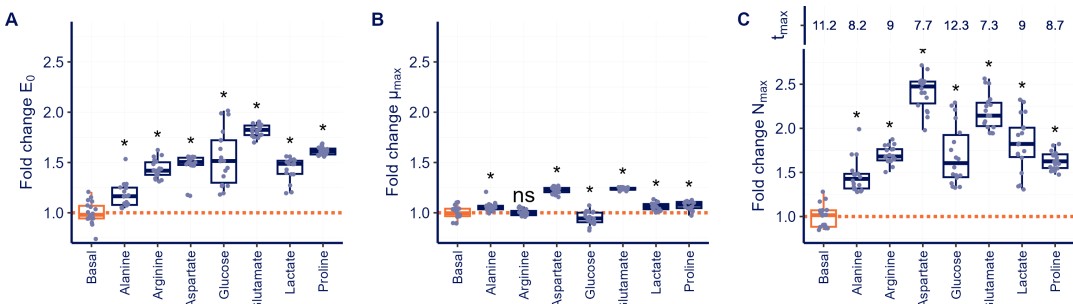

FIG 4 Nutrient effects on fitness and growth yield under antibiotic-free conditions. Growth curves for *P. aeruginosa* were analyzed for different media enriched for alanine, arginine, aspartate, glucose, glutamate, lactate, and proline on the fold change compared with basal media. (A) Total growth yield was described using the upper limit of the antibiotic concentration-response curve (E0), (B) The maximal growth rate ($\mu_{max}$) of the growth curve, (C) maximal population density $N_{max}$), and the time ($t_{max}$) required to reach $N_{max}$ are shown. Significant changes compared with the basal control media are indicated using '*' for $P < 0.05$ and "ns" for $P > 0.05$.

We re-fitted the PD model (Equation 1) with the *in vitro*-obtained growth and kill rates from our luminescence time course data per antibiotic concentration. In the basal media enriched with glucose and lactate, maximum bacterial growth rates were similar (0.25 $h^{-1}$ and 0.24 $h^{-1}$, respectively), as were the maximum bacterial kill rates (−0.15 $h^{-1}$ and −0.14 $h^{-1}$, respectively) (Fig. S6). However, the PD model estimated a 6-fold difference in the $EC_{50}$ for glucose-enriched (1.4 µg/mL) and lactate-enriched (8.6 µg/mL) environments, indicating that tobramycin is profoundly more effective at lower concentrations in glucose-rich culture conditions.

We simulated clinical tobramycin concentration-time profiles using a previously published PK model for an intravenous dose of 3.3 mg/kg administered every 8 h (Fig. 5A). The tobramycin PK simulation shows that the free drug concentrations fell below the $EC_{50}$ within 1 h for glucose-rich conditions and within 5.5 h for lactate-rich conditions after dose administration. As a result, treatment failure was observed for tobramycin under lactate-rich conditions, whereas growth suppression occurred in simulated glucose-enriched conditions (Fig. 5B).

## DISCUSSION

In this study, we used a combination of *in vitro* time-kill studies and mathematical modeling to investigate how specific nutrient conditions can distinctly affect bacterial growth and pharmacodynamic response of *P. aeruginosa* to different antibiotics.

We found that colistin, ciprofloxacin, imipenem, and tobramycin demonstrated >2-fold differences in nutrient-dependent changes in antibiotic sensitivity ($EC_{50}$), whereas these nutrients only had a limited effect on changes in bacterial fitness. Our time-course analysis revealed that changes in growth dynamics induced by these antibiotics occur within the initial hours of treatment, even when nutrients are abundant, and growth rates appear unchanged. This observation challenges the suggestion that antibiotic sensitivity changes were caused by nutrient depletion or diminished growth rates (30). In contrast, the response to aztreonam and ceftazidime under various nutrient conditions was more complex, as both the baseline response ($E_0$) and the maximum antimicrobial effect ($E_{max}$) were differently affected by the various nutrients.

Our findings indicate that adding glucose to nutrient-limited media enhances colistin sensitivity. Variations in colistin sensitivity under different nutrient conditions are thought to arise from nutrient-induced changes in cell wall structure (14, 15). Glucose-rich conditions have been previously suggested to decrease colistin sensitivity by stabilizing intracellular osmotic pressure (14). Our finding of enhanced colistin sensitivity thus challenges the hypothesis of osmotic stabilization of glucose in nutrient-scarce conditions. This observation is consistent with documented increases in colistin sensitivity in minimal media supplemented with glucose (31).

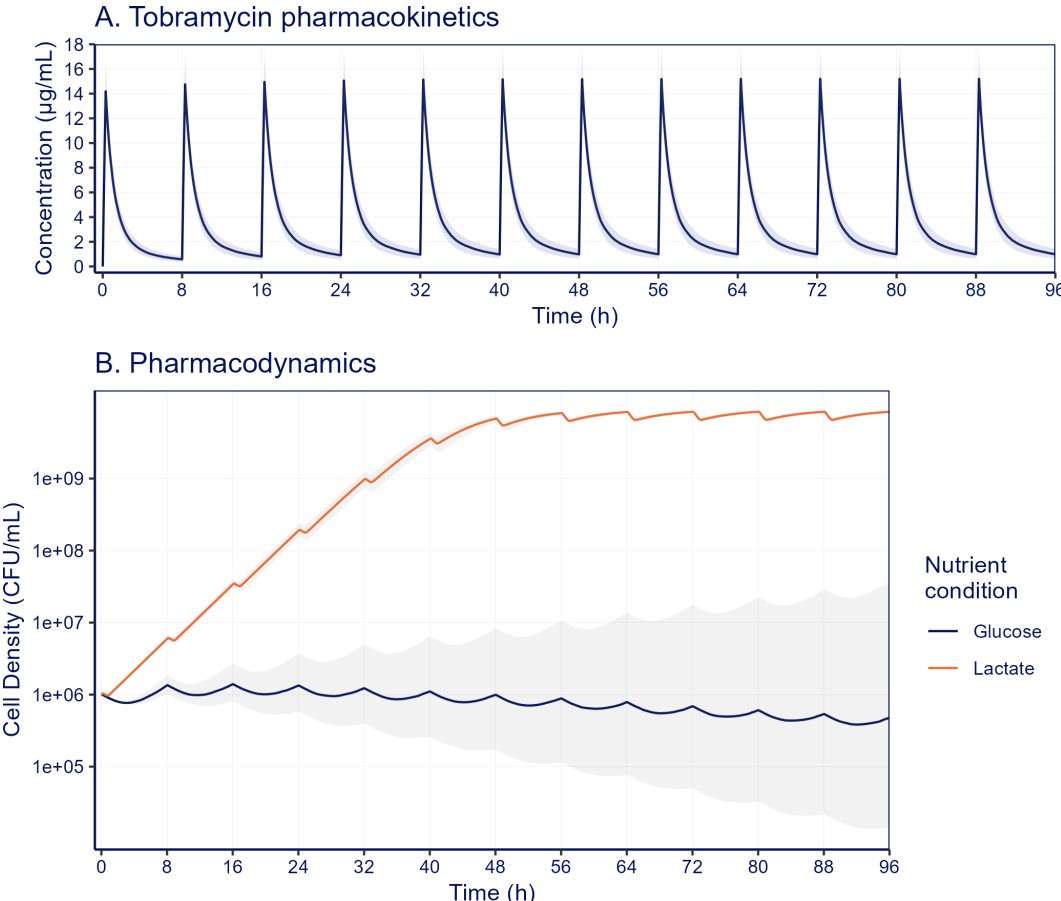

**FIG 5** Pharmacokinetic and pharmacodynamic simulation of tobramycin treatment in glucose or lactate-rich environments. (A) Tobramycin concentrations are modeled using a two-compartment model following a 3.3 mg/kg q8h dosing regimen. (B) Treatment response is simulated using a pharmacodynamic model based on population growth rates per drug concentration from *in vitro* growth/kill curves. The solid lines represent the median (1,000 simulations) with the interquartile range represented by the transparent-hued areas.

We found a diminished sensitivity of imipenem under nutrient conditions involving arginine, aspartate, glutamate, or proline. This can be explained by reduced imipenem uptake due to porin competition with these amino acids. Indeed, imipenem susceptibility in *P. aeruginosa* relies on the presence of outer membrane porins, particularly OprD and OprP, which facilitate the diffusion of sugars and amino acids (32–34). Furthermore, nutrient starvation upregulates OprD (32, 33, 35), providing an explanation for the increased imipenem sensitivity observed in both basal and glucose-rich media. The reduced growth rate and short exponential growth phase in these conditions may prompt an earlier starvation response, thereby enhancing OprD-mediated imipenem uptake.

We observed reduced ciprofloxacin susceptibility in glutamate media, which has previously been associated with adaptations in nitrogen metabolism and stress responses (36, 37). This metabolic adaptation mitigates ciprofloxacin's antibacterial effect of inducing oxidative stress by increasing the generation of reactive oxygen species during oxidative phosphorylation (38, 39). The increased ciprofloxacin sensitivity observed in arginine-rich conditions may be attributed to the induction of biofilm formation during treatment. Arginine-induced biofilm formation imposes a high metabolic burden on the cells (40), aligning with the effective anti-biofilm activity of ciprofloxacin (41). The difference in ciprofloxacin susceptibility among nutrient conditions might be due to a pH-dependent effect, although our medium was phosphate buffered to a pH of 7.4. Our observations in ciprofloxacin susceptibility correspond

to previous findings of ciprofloxacin being more effective in alkaline conditions, for example, arginine, compared with less sensitivity in acidic conditions, for example, glutamate and aspartate (42). However, this pH-mediated effect is not present in the observed reduced tobramycin susceptibility in arginine-rich conditions. Unbuffered arginine increases media alkalinity, resulting in increased tobramycin cellular uptake by increasing the transmembrane potential (43).

In our study, for tobramycin, we observed enhanced sensitivity for proline and glucose, whereas for lactate and alanine, reduced sensitivity was found. So far, previous studies have only investigated the effect of glucose-enriched media on *P. aeruginosa* tobramycin sensitivity, finding a similar potentiation effect (17, 44). Cellular respiration is key for aminoglycoside uptake, thereby directly relating tobramycin susceptibility to energy metabolism (17). The nutrients alternated in our media compositions are all closely linked to the TCA cycle, and intermediate products have been consistently correlated with tobramycin potentiation (17, 18, 44, 45). Interestingly, the sensitivity enhancement associated with TCA cycle activity can be suppressed by reducing the production of electron carriers through the activation of pleiotropic metabolic pathways. The redox imbalance induced by these alternative pathways and anaerobic energy production can be mitigated through the utilization of lactate (46). This observation may provide an explanation for the reduced susceptibility in lactate-rich media. Although proline and alanine demonstrated a profound effect on tobramycin treatment in our study, and previous research highlighted their role in alternative energy-producing pathways such as denitrification (47, 48), their exact role in *P. aeruginosa* metabolism during tobramycin treatment remains to be investigated.

Our PK-PD simulation illustrates how differences in PD response under nutrient-enriched conditions may lead to clinically relevant changes in antibiotic treatment response. This is demonstrated using a clinical tobramycin PK profile and the PD parameters from glucose and lactate-enriched conditions. Although these *in vitro* conditions do not fully replicate *in vivo* growth environments, which may also involve phenotypical adaptations such as biofilm formation or interspecies interactions, they underscore the relevance of considering nutrient conditions in the infectious microenvironment. This is especially relevant when nutrient availability could be altered under specific disease conditions. For instance, elevated lactate levels have been found in CF patients with declining lung function (49), which could thus potentially contribute to the reduced tobramycin efficacy in adult CF patients (50). Diabetes is a common disease in CF patients, for which increased glucose levels can be expected, which could potentially affect TOB treatment response (51).

The nutrient conditions employed in this study do not capture the full complexity of potential CF lung environments but provide isolated insights into the effect of specific nutrient conditions. Nutrients showed a modest differential impact on bacterial fitness ($\mu_{max}$) and profound changes in growth yield ($N_{max}$). The minimal impact on $\mu_{max}$ from substituting a single nutrient is consistent with prior studies on glucose and lactate addition to minimal media (52) and can be explained by a compressed nutrient utilization hierarchy under nutrient-poor conditions (53, 54), facilitating the simultaneous utilization of the basal medium nutrients and the added nutrients. This efficient metabolic regulation of *P. aeruginosa* suggests that our findings may not directly extrapolate to other conditions or nutrient combinations. Future research, focusing specifically on nutrient utilization during antibiotic exposure, will be crucial to deepen our understanding of specific nutrients' roles in more complex environments.

In conclusion, our study demonstrates a profound impact of specific nutrient conditions on antibiotic sensitivity, with only modest effects on fitness. Although broader clinical applicability of our results remains to be further elucidated, our work underscores the relevance of nutrients in the infectious microenvironment. Ultimately, it could be envisioned that specific nutrient levels in either plasma or sputum may be considered a clinically relevant predictor of antibiotic treatment response. Similarly, the

effect of nutrient conditions may be important for consideration in antibiotic susceptibility testing.

## ACKNOWLEDGMENTS

We wish to acknowledge Anh Duc Pham for his support in implementing the PK-PD simulations, and Suruchi Nepal for her critical review of the manuscript. All authors declare no competing interests.

## AUTHOR AFFILIATION

[1]Leiden Academic Centre for Drug Research, Leiden University, Leiden, the Netherlands

## AUTHOR ORCIDs

Maik Kok  http://orcid.org/0000-0002-7331-7549
J. G. Coen van Hasselt  http://orcid.org/0000-0002-1664-7314

## AUTHOR CONTRIBUTIONS

Maik Kok, Conceptualization, Data curation, Formal analysis, Investigation, Methodology, Validation, Visualization, Writing – original draft | Thomas Hankemeier, Funding acquisition, Supervision | J. G. Coen van Hasselt, Conceptualization, Funding acquisition, Investigation, Methodology, Supervision, Writing – review and editing

## ADDITIONAL FILES

The following material is available online.

### Supplemental Material

**Supplemental material (Spectrum01409-24-S0001.pdf).** Figures S1 to S6; Tables S1 to S3.

### Open Peer Review

**PEER REVIEW HISTORY (review-history.pdf).** An accounting of the reviewer comments and feedback.

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
