## [Reviewer comments · Microbiology Spectrum]

Microbiology Spectrum

Nutrient conditions affect antimicrobial pharmacodynamics in *Pseudomonas aeruginosa*

Maik Kok, Thomas Hankemeier, and J. G. Coen van Hasselt

Corresponding Author(s): J. G. Coen van Hasselt, Universiteit Leiden

Review Timeline:

Submission Date:	June 15, 2024
Editorial Decision:	September 1, 2024
Revision Received:	October 25, 2024
Accepted:	November 8, 2024

Editor: Silvia Cardona

Reviewer(s): Disclosure of reviewer identity is with reference to reviewer comments included in decision letter(s). The following individuals involved in review of your submission have agreed to reveal their identity: Cheryl L. Quinn (Reviewer #1)

Transaction Report:

DOI: <https://doi.org/10.1128/spectrum.01409-24>

Re: Spectrum01409-24 (Nutrient conditions affect antimicrobial pharmacodynamics in *Pseudomonas aeruginosa*)

Dear Prof. J. G. Coen van Hasselt:

Thank you for submitting your manuscript to Microbiology Spectrum. Our policy for publication is to accept manuscripts that are technical and methodologically sound. Reviewers of your manuscript agree that the topic of how nutrients affect antibiotic susceptibility and how this can partially explain inconsistencies between standard susceptibility tests and treatment outcomes is very relevant to the field. However, reviewers have noted methodological shortcomings, specifically related to the RLU signal as a readout for antibiotic susceptibility. Reviewers have concerns that this signal may be affected directly but the media conditions and pH, which may turn into a less clear correlation with bacterial growth and the standard MIC. I realize this is a major concern for the work and may require additional experiments and controls.

If you feel you can address the reviewers' concerns, I will be willing to consider a revised manuscript. Reviewers' comments are below.

Revision Guidelines

Sincerely,
Silvia Cardona
Editor
Microbiology Spectrum

Reviewer #1 (Comments for the Author):

This is a really important study that triggers insight on selection of antibiotics and dosing to treat bacterial infections. We know that the nutrient environment impacts sensitivity of bacteria to antibiotics, and we know different patient populations have different levels of sugar in tissues and blood. While it may be hard to test nutrient levels in tissues at the very least this approach could help guide treatment of blood infections.

In general, the paper is well written and the studies performed are thorough and well controlled. The data is summarized nicely in their figures. The supplementary figures are very useful in evaluating the rigor of their studies.

The biggest concern about the study design is the potential for pH differences in the media they use. They start with M9 media that is phosphate buffer. But they are adding various acids, bases or buffers in their different media tests. We need to know that they checked the pH levels and they are all the same across the media. pH levels greatly impact the sensitivity of bacteria to antibiotics. Can they confirm this was done? They may argue that if you do the pH calculations for phosphate buffered media that the additions should not change the pH. I would argue that they need to confirm this as this is also a way to confirm their media were made correctly.

If they did this pH test, they can add that line to their Materials & Methods section.

If they have not checked the pH levels they need to do so before publishing.

Reviewer #2 (Comments for the Author):

The study by Kok et al. examined the susceptibility of *Pseudomonas aeruginosa* to several antibiotics and its fitness under different nutrient conditions. The authors identified some alterations both in fitness and antibiotic response - several of which are in agreement with previous reports. This work also included a simulation predicting the PK-PD profile of antibiotics under the tested conditions in an attempt to provide insights into the potential impact of the in vitro phenotypes on the therapeutic outcome. Indeed, there is increasing evidence and proof-of-concept studies suggesting the potential importance of the growth conditions and nutrients at the site of infections on the antibiotic therapeutic outcome. Therefore, additional studies, such as this one, further characterizing the effects of nutrients are of interest and potential clinical relevance. However, there are several major concerns in this study as described below.

Major Comments:

1. Antibiotic susceptibility testing and definitions: the choice of assay format - a bioluminescence-based assay - and the readouts (EC50, E0, etc.) are unconventional and would not be easily interpreted and compared across other studies in the literature.

A) Specifically, all AST testing in this study is done in vitro in supplemented M9 media in which direct turbidimetric growth measurements are feasible. The rationale behind using a bioluminescent PA01 strain is not clear. A CFU or OD to RLU (and RLU AUC) correlation in minimal media in the presence of each carbon source should be assessed to determine whether the observed RLU differences (including maximal signals in each C source) are due to altered physiological responses or artifacts of lux activity or signal in each of these nutrient conditions.

B) More commonly used readouts, especially MIC values, should be reported. MIC shifts are easier to interpret in the field. Also, some definitions are not clear. For example, how the relative and absolute EC50 are defined and the relevance and interpretation of any difference thereof (e.g., in Lines 209-219) are not clear. Also, there should be further elaboration on how those interpretations are made and whether these interpretations are supported by previous literature.

2. Fitness calculations in Figure 4 and the associated analyses: The raw data and growth controls used to derive the results shown in the figure and similar assays not based on RLU measurements should be provided to validate these results. Notably, the results shown in Figure 2 shows a starting AUC (RLU) level (what is considered E0 by the authors) in the absence of antibiotics that is relatively inconsistent for the same nutrient/medium across all plots. For e.g., compare the baseline of each medium in 2C (relatively more clustered) vs 2E (separated). The biological relevance of the differences of that magnitude on the RLU scale are hard to assess - (related to comment 1: these would be better assessed with a CFU or OD correlation). These observations suggest a validation of the ratios shown in Figure 4 is required.

3. PK-PD simulations (Lines 221-241): The title of the section (Line 221) is ambiguous and may imply that this section directly assesses the treatment responses experimentally rather than in a simulation; hence should be revised. Some sentences also imply that bacterial growth was determined experimentally (e.g., Line 240); these should also be clarified. Importantly, the validity and limitations of the simulations as a proof-of-concept for in vivo treatment should be discussed. Specifically, the simulation uses PD data derived from the M9 media supplemented with a specific carbon source which do not match those from which the PK data are derived. It is not clear how these simulations are predictive of the patient outcomes.

Other comments

a) Lines 60-72: A brief discussion on the similarities/differences between nutrient depleted conditions discussed in this

paragraph and CF sputum described in the previous sections should be included. CF sputum could be considered nutrient rich and typically includes a large array of carbon sources. The narrative might imply that CF sputum is nutrient limited.

b) Lines 74-78: The rationale for selecting these antibiotics and carbon sources should be discussed.

c) Abbreviations such as PK-PD should be defined on first mention.

d) Lines 176-191: The flow of this section is challenging to follow. The discussion jumps between different antibiotics and nutrients.

e) Lines 210-214: Absolute EC50 was frequently used in the previous section without being defined. The explanation for what EC50 means should be placed earlier to aid in clarity when discussing results.

f) Line 238: the description of the data is a bit confusing.

g) Lines 252-253: This statement is ambiguous and may not necessarily be supported by the results

h) Line 279: Ciprofloxacin is misspelled

Reviewer 1:

This is a really important study that triggers insight on selection of antibiotics and dosing to treat bacterial infections. We know that the nutrient environment impacts sensitivity of bacteria to antibiotics, and we know different patient populations have different levels of sugar in tissues and blood. While it may be hard to test nutrient levels in tissues at the very least this approach could help guide treatment of blood infections.

In general, the paper is well written and the studies performed are thorough and well controlled. The data is summarized nicely in their figures. The supplementary figures are very useful in evaluating the rigor of their studies.

Response: We would like to thank the reviewer 1 for their kind words about our manuscript and are pleased to see that the reviewer recognizes the relevance of our study.

1. The biggest concern about the study design is the potential for pH differences in the media they use. They start with M9 media that is phosphate buffer. But they are adding various acids, bases or buffers in their different media tests. We need to know that they checked the pH levels and they are all the same across the media. pH levels greatly impact the sensitivity of bacteria to antibiotics. Can they confirm this was done? They may argue that if you do the pH calculations for phosphate buffered media that the additions should not change the pH. I would argue that they need to confirm this as this is also a way to confirm their media were made correctly.

If they did this pH test, they can add that line to their Materials & Methods section.

If they have not checked the pH levels they need to do so before publishing.

Response: We agree with the reviewer's suggestion to be more specific about the role of pH. To clarify, the pH of both the basal medium and the nutrient-specific media were confirmed and theoretical calculations regarding the buffer capacity suggest only minor changes in pH < 0.1. We have made the following changes in the manuscript [Line 84-87]:

A basal medium was prepared consisting of physiologically relevant concentrations of amino acids in synthetic CF sputum as described previously (7), calcium and magnesium adjusted 0.1 M phosphate buffer, ammonium chloride, ferrous sulfate, Basal Medium Eagle 1x vitamins, and trace metals. The pH of the basal medium was confirmed to be 7.4, and was re-confirmed after addition of nutrients and filter sterilization.

Reviewer 2:

The study by Kok et al. examined the susceptibility of *Pseudomonas aeruginosa* to several antibiotics and its fitness under different nutrient conditions. The authors identified some alterations both in fitness and antibiotic response - several of which are in agreement with previous reports. This work also included a simulation predicting the PK-PD profile of antibiotics under the tested conditions in an attempt to provide insights into the potential impact of the in vitro phenotypes on the therapeutic outcome. Indeed, there is increasing evidence and proof-of-concept studies suggesting the potential importance of the growth conditions and nutrients at the site of infections on the antibiotic therapeutic outcome. Therefore, additional studies, such as this one, further characterizing the effects of nutrients are of interest and potential clinical relevance. However, there are several major concerns in this study as described below.

Response: We are pleased that reviewer 2 recognizes the importance of studies highlighting the role of nutrients at the site of infection. Based on the questions of this reviewers, we conclude that due to unclarities in the wording of our manuscript, several misconceptions may have occurred, which we now tried to improve accordingly.

- 1) Antibiotic susceptibility testing and definitions: the choice of assay format - a bioluminescence-based assay - and the readouts (EC50, E0, etc.) are unconventional and would not be easily interpreted and compared across other studies in the literature.

Response: Luminescence-based assays are indeed less conventional, but this choice was made deliberately by us. The use of this luminescent reporter strains has enabled us to execute high-throughput time kill studies, which in contrast to static MIC testing or OD-based growth assays, can fully quantify the time course of drug effects including bacterial kill. Furthermore, our subsequent pharmacodynamic analysis quantifying specific PD parameters, further provides enhanced insight into specific antimicrobial pharmacodynamic parameters. We have added a new paragraph in the Introduction section to address this point accordingly, as follows:

New paragraph added [72-73]:

To assess the effects of nutrient conditions on antimicrobial pharmacodynamics (PD), conventional readouts such as the minimum inhibitory concentrations (MIC) have important limitations, as this is a static composite measure. More comprehensive characterization of changes in the pharmacodynamic response to antibiotics can be achieved through time kill studies, which monitor bacterial densities over time when exposed to antibiotics, allowing the evaluation of bacterial growth, antibiotics-associated killing, and adaptation effects (22, 23). Although time-kill studies provide these valuable insights, they remain limited in their throughput and the number of time points at which data can be collected (24). The use of bacterial strains carrying luminescent reporters allows real-time monitoring of bacterial growth and killing dynamics during antibiotic exposure (25, 26). The resulting profiles can be analyzed using mathematical pharmacodynamic models to obtain further quantitative insights into PD relationships. As such, the use of luminescence-based time kill studies in combination with quantitative pharmacodynamic models is well-suited for comprehensively assessing the effects of nutrient conditions on antibiotic response.

Revisions to final Introduction paragraph [78-81]:

*We assessed the bacterial growth/kill time course profiles using extensive time-kill studies with a modified *P. aeruginosa* PAO1 strain carrying a constitutively active luminescent*

reporter. This strain was subsequently used to infer PD parameters and perform pharmacokinetic-pharmacodynamic (PK-PD) simulations to demonstrate the potential clinical impact of nutrient on antimicrobial PD.

Revision to the Material and Methods section [119-120]:

The P. aeruginosa bioluminescent strain PAO1 Xen41 was used in all experiments. The promoterless insertion of the luxCDABE cassette into the chromosomal genome resulted in a linear relationship between luminescence in relative light units (RLU) and CFU/mL (Figure S1) (25-26).

References to real-time monitoring of genetically modified strains:

25. H. L. Rocchetta (2001). *Antimicrobial Agents and Chemotherapy*, 45(1), 129-137
26. G. R. Siragusa (1999). *Applied and Environmental Microbiology*, 65(4), 1738–1745

- 2) Specifically, all AST testing in this study is done in vitro in supplemented M9 media in which direct turbidimetric growth measurements are feasible. The rationale behind using a bioluminescent PAO1 strain is not clear. A CFU or OD to RLU (and RLU AUC) correlation in minimal media in the presence of each carbon source should be assessed to determine whether the observed RLU differences (including maximal signals in each C source) are due to altered physiological responses or artifacts of lux activity or signal in each of these nutrient conditions.

Response: This study does not perform antimicrobial susceptibility testing, but rather characterized the full pharmacodynamic response to antibiotics under different media conditions. Regarding the rationale for using RLU, we refer to our response to the previous response (point 1).

We are confident that RLU well reflects underlying CFU counts. We have added a new figure (**Figure S1**, inserted at the end of the rebuttal letter), which demonstrate a linear relationship between RLU and CFU/mL.

OD time course data is not available, and in our opinion is not needed to support the conclusions drawn in the current analysis.

The bioluminescent reporter strain used contains a constitutively active transcribed lux cassette reporter. The luminescent signal produced by the proteins encoded by the lux operon is independent of extracellular components. Additionally, there is a direct correlation between the colony-forming unit (CFU) counts of the inoculum and the starting relative luminescence units (RLU), with similar growth patterns observed across different growth conditions during the initial hours. This indicates consistent luminescent activity across conditions.

- 3) More commonly used readouts, especially MIC values, should be reported. MIC shifts are easier to interpret in the field.

Response: Please refer to our response to point 1 for a more extensive response and changes made to the manuscript regarding MIC and our choice for RLU. Although we agree MIC is a conventional measure for antibiotic susceptibility, the purpose of the current analysis was to further characterize the full bacterial pharmacodynamics, including kill effects. We have not measured changes in MIC in this study.

- 4) Also, some definitions are not clear. For example, how the relative and absolute EC₅₀ are defined and the relevance and interpretation of any difference thereof (e.g., in Lines 209-219) are not clear.

Response: We agree that our statements about absolute and relative EC₅₀ were too ambitious [lines 209-219]. Changes between absolute and relative EC₅₀ can be influenced by several other factors such as internal antibiotic potency and require more in depth study. We removed our claims which directly correlate bacterial fitness and changed it to theoretical statements demonstrating the differences between antibiotics [lines 209-219], as follows:

*The impact of differences in E_0 across different nutrient conditions on PD parameters was further evaluated by analyzing the total antimicrobial response. Comparing the relative EC₅₀ with the absolute EC₅₀ provides an indication how the limits of PD model influence the total antimicrobial effect. The relative EC₅₀ is defined as the midpoint between the two limits of concentration-effect curve, whereas the absolute EC₅₀ denotes a 50% reduction in the AUC from the baseline with no antimicrobial effect (E_0). A larger discrepancy between these EC₅₀ values suggests a stronger impact of the two limits on determining the antibiotic EC₅₀ (31). For treatments with ciprofloxacin, colistin, imipenem and tobramycin, the difference between the average relative and absolute EC₅₀ values was less than 5% (**Figure S4**). In contrast, ceftazidime and aztreonam treatments showed difference of respectively 14% and 22% indicating that differences in the PD model limits between the nutrient conditions do influence the determination of EC₅₀.*

We improved definitions of PD parameters at the start of the results section [lines 161-171], as follows,

*We cultured *P. aeruginosa* under various nutrient conditions in the presence of different antibiotics to investigate the effect of nutrients on the pharmacodynamic (PD) response. To summarize the bacterial response kinetics—encompassing growth enhancement, suppression or killing during antibiotic treatment—we calculated the AUC of the luminescence time course profiles. We then regressed the AUC values against antibiotic concentrations using a sigmoidal Emax model, allowing us to visualize differences in the PD response across conditions (**Figure 2**) (18). Overall, these analyses revealed significant effects of nutrients on the antibiotic concentration required to achieve 50% of the total antimicrobial effect (relative EC₅₀), and the steepness of the concentration-response profiles.*

The relative EC₅₀ would be the primary metric of relevance to quantitatively indicate subtle changes in drug potency, i.e., antibiotic sensitivity across conditions.

- 5) Also, there should be further elaboration on how those interpretations are made and whether these interpretations are supported by previous literature.

Response: To address this point we have in our response to comment 1 included now a brief discussion and supporting literature citations on the benefits of full PD models over MIC values (17) and the use of PD models to describe changes in time-kill data (18). In the response to point 4 we also included a new reference describing the use of relative and absolute EC₅₀ estimates in Emax models (21).

24. R. Regoes (2004). Antimicrobial agents and Chemotherapy, 48, 3670-3676.

31. Z. Noel (2017). Plant disease, 102(4), 708-714.

- 6) Fitness calculations in Figure 4 and the associated analyses: The raw data and growth controls used to derive the results shown in the figure and similar assays not based on RLU measurements should be provided to validate these results.

Response: The data used to derived growth rate (μ_{max}), maximal population (N_{max}), and the time to reach maximal population (t_{max}) were obtained from **Figure S2 (Figure S3** in the revised manuscript). To further support data transparency regarding parameters extracted from our models, we have included an additional figure (**Figure S4**, located at the end of the rebuttal letter)

- 7) Notably, the results shown in Figure 2 shows a starting AUC (RLU) level (what is considered E_0 by the authors) in the absence of antibiotics that is relatively inconsistent for the same nutrient/medium across all plots. For e.g., compare the baseline of each medium in 2C (relatively more clustered) vs 2E (separated). The biological relevance of the differences of that magnitude on the RLU scale are hard to assess - (related to comment 1: these would be better assessed with a CFU or OD correlation). These observations suggest a validation of the ratios shown in Figure 4 is required.

Response: In our view, the variation between replicates in antibiotic-free conditions is low, as shown in **Figure S2** and **Figure 4 panels B and C**. The variation observed in **Figure 2** arises from small differences in the model fit at lower antibiotic concentrations, resulting in minor discrepancies between the AUC in antibiotic-free conditions and the baseline E_0 estimate. As a result, E_0 can vary slightly within nutrient conditions between experiments with different antibiotics. The new supplemental figure (**Figure S3**), discussed in our response to Comment 4, provides further insight into the source of E_0 variation (see **Figure 4 panel A**). We changed Figure 2 to a fixed y-scaling to improve the observation of differences within conditions (see end of the rebuttal letter).

The relevance of this variation in the limits of our model for antibiotic sensitivity is assessed through its impact on EC_{50} determination, as discussed in our response to comment 3 regarding absolute and relative EC_{50} .

We acknowledge an error in lines 196-198 and the free y scaling of Figure 2 may cause confusion, which is revised as follows:

Except for alanine, for all nutrients we found an increase of >1.5 fold in the upper limit of the model (E_0), i.e., the response baseline with no antimicrobial induced reduction of the AUC (Figure 2).

- 8) PK-PD simulations (Lines 221-241): The title of the section (Line 221) is ambiguous and may imply that this section directly assesses the treatment responses experimentally rather than in a simulation; hence should be revised.

Response: This was not our intent. We revised the title to prevent misconceptions as follows [lines 221-222]:

In vitro nutrient-driven PD differences impact treatment simulations with a clinically relevant tobramycin PK profile.

- 9) Some sentences also imply that bacterial growth was determined experimentally (e.g., Line 240); these should also be clarified. Importantly, the validity and limitations of the simulations as a proof-of-concept for in vivo treatment should be discussed.

Response: We have revised the opening sentence to clarify the rationale behind our simulation [lines 223-225], as follows:

To assess whether the magnitude of nutrient-associated changes in the PD response observed in vitro may have significance at clinically relevant antibiotic concentrations, we performed pharmacokinetic-pharmacodynamic (PK-PD) simulations.

We have revised the sentence introducing the PD model for the simulation [lines 227-229], as follows:

*We re-fitted the PD model (**Equation 1**) with the in vitro obtained growth and kill rates from our luminescence time course data per antibiotic concentration.*

We have revised the description of the media conditions [lines 229-232], as follows:

*In the basal media enriched with glucose and lactate, maximum bacterial growth rates were similar (0.25 h^{-1} and 0.24 h^{-1} , respectively), as were the maximum bacterial kill rates (-0.15 h^{-1} and -0.14 h^{-1} , respectively) (**Figure S4**).*

We revised the unclear lines [237-241] as suggested by the reviewer as follows:

*The tobramycin PK simulation shows that the free drug concentrations fell below the EC_{50} within 1 hour for glucose-rich conditions and within 5.5 hours for lactate-rich conditions after dose administration. As a result, treatment failure for tobramycin under lactate-rich conditions, while growth suppression was observed in simulated glucose-enriched conditions (**Figure 5B**).*

- 10) Specifically, the simulation uses PD data derived from the M9 media supplemented with a specific carbon source which do not match those from which the PK data are derived. It is not clear how these simulations are predictive of the patient outcomes.

Response: Changes in antibiotic PD parameters varied in magnitude. As such, we performed PK/PD simulations to put these changes into some clinical context. We of course fully agree that our supplemented media does not mimic in vivo growth conditions. In addition to the revisions made in the results section described in our response to reviewer comment 10, we have also revised the sentence in the abstract that refers to our PK-PD simulations [lines 30-32], as follows:

Using pharmacokinetic-pharmacodynamic simulations, we illustrate that the magnitude of the nutrient-driven pharmacodynamic changes impact treatment.

We have revised the closing sentence of the introduction [lines 223-225], as follows:

The impact of these conditions on treatment with clinically relevant dynamic antibiotic concentrations was further illustrated by performing pharmacokinetic-pharmacodynamic (PK-PD) simulations.

We have revised the paragraph in the discussion section regarding the PK-PD simulation results [lines 309-312], as follows:

Our PK-PD simulation illustrate how differences in PD response under nutrient-enriched conditions may lead to clinically relevant changes in antibiotic treatment response. This is demonstrated using a clinical tobramycin PK profile and the PD parameters from glucose and lactate-enriched conditions. While these in vitro conditions do not fully replicate in vivo growth environments, which may also involve phenotypical adaptations such as biofilm formation or interspecies interactions, they underscore the relevance of considering nutrient conditions in the infectious microenvironment. This is especially relevant when nutrient availability could be altered under specific disease conditions.

11) Lines 60-72: A brief discussion on the similarities/differences between nutrient depleted conditions discussed in this paragraph and CF sputum described in the previous sections should be included. CF sputum could be considered nutrient rich and typically includes a large array of carbon sources. The narrative might imply that CF sputum is nutrient limited.

Response: We agree that CF sputum may be considered nutrient-rich and that these sentences can be interpreted differently. We removed the reference to nutrient deprived environments in line 62-63. We added this line to the new introduction paragraph mentioned in response to comment 1. We further revised the paragraph referred to in this comment.

Alterations in metabolic processes associated with differences in available nutrients may impact response to antibiotic treatment in *P. aeruginosa* (11–13). For example, nutrient deprivation prevents cell wall modifications due to its high energy demand, enhancing the effect of cell wall targeting antibiotics (e.g., polymyxins and β -lactams) (16–18). The supplementation of metabolites to activate energy production through aerobic respiration in nutrient deprived environments can increase sensitivity towards fluoroquinolones and aminoglycosides (22–24). While these changes illustrate the modulatory role of deprived nutrient conditions and microbial metabolism on the response to antibiotics, insights into the contribution of nutrients relevant to the CF lung microenvironments remain limited.

12) Lines 74-78: The rationale for selecting these antibiotics and carbon sources should be discussed.

Response: Revisions have been made to the introduction, addressing this point, as follows:

13) *In the current study, we aimed to systematically evaluate the impact of a wide range of CF sputum-relevant carbon sources on antimicrobial time-kill responses in *P. aeruginosa*. The nutrients evaluated included alanine, arginine, aspartate, glutamate, lactate, proline, and glucose. These nutrient-associated effects were evaluated for six antibiotics*

commonly used for respiratory tract infections in CF, including aztreonam, ceftazidime, ciprofloxacin, colistin, imipenem, and tobramycin. Abbreviations such as PK-PD should be defined on first mention.

Response: Corrected.

14) Lines 176-191: The flow of this section is challenging to follow. The discussion jumps between different antibiotics and nutrients.

Response: Revisions have been made to improve readability, as follows:

*When comparing the relative change in EC_{50} to the basal medium (**Figure 3B**), both aztreonam and ceftazidime exhibited similarly enhanced sensitivity across different nutrient conditions. The most notable changes were the increased sensitivity observed in lactate-enriched media for both antibiotics. In contrast, imipenem sensitivity was consistently reduced in all nutrient-enriched conditions, with the most significant reductions observed in aspartate- and glutamate-enriched media.*

For ciprofloxacin, colistin and tobramycin a wider variation in effect was compared to the basal medium. Glucose- and proline-enriched media resulted in a reduction of EC_{50} , while aspartate-, glutamate- and lactate-enriched media increased the EC_{50} for all three antibiotics. The largest change in sensitivity was observed for tobramycin, where for lactate-rich media, the EC_{50} value increased profoundly ($\log_2(FC_{EC50}) = 2.09$, a 4.4-fold increase).

15) Lines 210-214: Absolute EC_{50} was frequently used in the previous section without being defined. The explanation for what EC_{50} means should be placed earlier to aid in clarity when discussing results.

Response: Corrected.

16) Line 238: the description of the data is a bit confusing.

Response: Corrected.

17) Lines 252-253: This statement is ambiguous and may not necessarily be supported by the results

Response: Corrected.

18) Line 279: Ciprofloxacin is misspelled

Response: Corrected.

Revised Figures:

Figure 2. No revisions to caption.

New supplemental figures

Supplemental Figure 1. Linear calibration between luminescence (relative light units, RLU) and cell counts (CFU/mL) for multiple combinations of detector settings, varying iteration time (iter, columns) and gain (rows). The iteration time stands for the total measurement time per well and the gain is amplification in the conversion from light into electric signal.

Supplemental Figure 3. Emax model fitting was performed on the area under the curve (AUC) of growth curves across varying antibiotic concentrations. The model was fitted using the average AUC values for each antibiotic concentration ($n = 3$). From this model, the upper limit (E_0), the half-maximal effective concentration (EC_{50}), and the lower limit (E_{max}) were determined.

Re: Spectrum01409-24R1 (Nutrient conditions affect antimicrobial pharmacodynamics in *Pseudomonas aeruginosa*)

Dear Prof. J. G. Coen van Hasselt:

Your manuscript has been accepted, and I am forwarding it to the ASM production staff for publication. Your paper will first be checked to make sure all elements meet the technical requirements. ASM staff will contact you if anything needs to be revised before copyediting and production can begin. Otherwise, you will be notified when your proofs are ready to be viewed.

Sincerely,
Silvia Cardona
Editor
Microbiology Spectrum